# Outcome of Unilateral Pulmonary Edema after Minimal-Invasive Mitral Valve Surgery: 10-Year Follow-Up

**DOI:** 10.3390/jcm10112411

**Published:** 2021-05-29

**Authors:** Thomas Puehler, Christine Friedrich, Georg Lutter, Maike Kornhuber, Mohamed Salem, Jan Schoettler, Markus Ernst, Mohammed Saad, Hatim Seoudy, Derk Frank, Felix Schoeneich, Jochen Cremer, Assad Haneya

**Affiliations:** 1Department of Cardiac and Vascular Surgery, Campus Kiel, University-Medical-Center Schleswig-Holstein, Arnold-Heller-Str. 3, House C 2, D-24105 Kiel, Germany; christine.friedrich@uksh.de (C.F.); georg.lutter@uksh.de (G.L.); maikekornhuber@web.de (M.K.); Mohamed.Salem@uksh.de (M.S.); Jan.Schoettler@uksh.de (J.S.); markus.ernst@uksh.de (M.E.); felix.schoeneich@uksh.de (F.S.); Jochen.Cremer@uksh.de (J.C.); Assad.Haneya@uksh.de (A.H.); 2DZHK (German-Centre for Cardiovascular-Research), Partner Site Hamburg/Kiel/Lübeck, D-24105 Kiel, Germany; derk.frank@uksh.de; 3Department of Internal Medicine III, Cardiology and Angiology, Campus Kiel, University-Medical-Center Schleswig-Holstein, D-24105 Kiel, Germany; mohammed.saad@uksh.de (M.S.); hatim.seoudy@uksh.de (H.S.)

**Keywords:** minimally invasive, mitral valve surgery, unilateral lung edema, UPE, ECMO, ECLS

## Abstract

The study was approved by the institutional review board (IRB) at the University Medical Center Campus Kiel, Kiel, Germany (reference number: AZ D 559/18) and registered at the German Clinical Trials Register (reference number: DRKS00022222). Objective. Unilateral pulmonary edema (UPE) is a complication after minimally invasive mitral valve surgery (MIMVS). We analyzed the impact of this complication on the short- and long-term outcome over a 10-year period. Methods. We retrospectively observed 393 MIMVS patients between 01/2009 and 12/2019. The primary endpoint was a radiographically and clinically defined UPE within the first postoperative 24 h, secondary endpoints were 30-day and long-term mortality and the percentage of patients requiring ECLS. Risk factors for UPE incidence were evaluated by logistic regression, and risk factors for mortality in the follow-up period were assessed by Cox regression. Results. Median EuroSCORE II reached 0.98% in the complete MIMVS group. Combined 30-day and in-hospital mortality after MIMVS was 2.0% with a 95, 93 and 77% survival rate after 1, 3 and 10 years. Seventy-two (18.3%) of 393 patients developed a UPE 24 h after surgery. Six patients (8.3%) with UPE required an extracorporeal life-support system. Logistic regression analysis identified a higher creatinine level, a worse LV function, pulmonary hypertension, intraoperative transfusion and a longer aortic clamp time as predictors for UPE. Combined in hospital mortality and 30-day mortality was slightly but not significantly higher in the UPE group (4.2 vs. 1.6%; *p* = 0.17). Predictors for mortality during follow-up were age ≥ 70 years, impaired RVF, COPD, drainage loss ≥ 800 mL and length of ventilation ≥ 48 h. During a median follow-up of 4.6 years, comparable survival between UPE and non-UPE patients was seen in our analysis after 5 years (89 vs. 88%; *p* = 0.98). Conclusions. In-hospital outcome with UPE after MIMVS was not significantly worse compared to non-UPE patients, and no differences were observed in the long-term follow-up. However, prolonged aortic clamp time, worse renal and left ventricular function, pulmonary hypertension and transfusion are associated with UPE.

## 1. Introduction

There is a high demand for minimally invasive mitral valve surgery (MIMVS), and accordingly, the numbers of procedures in national databases are rising [1]. Though the current guidelines do not expressly recommend the surgical technique, the latter is often speculated to be the gold standard in the treatment of isolated mitral valve therapy [2]. The results of MIMVS are not considered inferior to conventional surgery, but there are some basic disadvantages associated with MIMVS compared to conventional procedures: one disadvantage is the highly specialized surgery [3,4]. Furthermore, some mitral valve pathologies with difficult reconstruction properties complicate access site conditions [5].

One crucial and yet unsolved aspect of MIMVS is the occasional postoperative unilateral pulmonary edema (UPE) first described in 2009 [6]. Some risk factors such as prolonged cardiopulmonary bypass time (CPB) and preoperative C-reactive protein as an indicator for high disposition for inflammatory response were described [7,8,9,10]. 

However, less is known about the short- and long-term survival after UPE. Therefore, we investigated the risk factors for a new onset of UPE and the impact of this complication on the short- and long-term outcome over a 10-year period.

## 2. Materials and Methods

Between July 2009 and June 2019, isolated MIMVS were conducted in 629 patients. Three hundred and ninety-three patients were included into this retrospective observational study. Patient details were collected from digital patient charts and electronical hospital information systems. The study was approved by the institutional review board (IRB) at the University Medical Center Campus Kiel (reference number: AZ D 559/18) and registered at the German Clinical Trials Register (reference number: DRKS00022222). Patient’s informed consent was waived for publication of this study.

## 3. Primary and Secondary Endpoints

The primary endpoint was defined as newly occurring UPE within the first postoperative 24 h. Thirty-day mortality, long-term mortality and percentage of patients requiring ECLS in patients after MIMVS were secondary endpoints. Risk factors for the incidence of UPE after MIMVS were evaluated in a multivariable logistic regression analysis. UPE was determined according to the diagnostic criteria of re-expansion pulmonary-edema [11] if more than 20% opafication of the right hemithorax was detected and no direct or indirect signs for atelectasis or other infiltrates occurred within the first 24 h postoperatively (Figure 1). Pulmonary edema was assessed by at least three chest X-rays (at admission to the ICU, after 6 h, within 24 h), and radiographs were evaluated by 2 independent radiologists using uniform criteria. Risk stratification was assessed by logistic European System for Cardiac Operative Risk Evaluation (EuroSCORE), as described by Roques et al. 2003 [12]. Pulmonary hypertension was scaled > or <60 mmHg according to the requested values in our institutional database.

## 4. Surgical Technique and Anesthetic Management

As recommended by the WHO, prior to surgery, a patient-specific team timeout protocol was filled out in the operation room. After bringing the patient firstly to a supine position with a right-lifted thorax and secondly placing an open access to transfemoral venous and arterial cannulation, extracorporeal circulation was established. Correct positioning of the venous cannula was guided by transesophageal echocardiography. In some cases, additional cannulation of the upper vena cava was previously established. During the main surgical procedure, body temperature was targeted to moderate hypothermia (28–32 °C). Access to the mitral valve was then performed using a right lateral mini thoracotomy through the 4th to 5th intercostal space, as described by Colvin and co-workers [13]. In addition to the antegrade application of cold Buckburg solution after aortic clamping, retrograde cardioplegia was additionally administered in 90% of patients. Carbon-dioxide was continuously insufflated in the hemithorax to minimize the risk of air embolism. Further specific techniques such as exposition of the operative situs, anesthetic intra and perioperative management and lung ventilation have been described by Renner and coworkers in 2016 [9]. 

## 5. Statistical Methods

Continuous variables were summarized as median with 25th and 75th percentiles and compared by the Mann–Whitney U test. Categorial data were presented as absolute (n) and relative (%) frequencies and compared by the Chi^2^ test or Fisher’s exact test, as appropriate. 

Survival patients were calculated on right-censored data by Kaplan–Meier analyses, presented as survival curves and compared for differences between subgroups by log-rank test. 

Age, gender, pre- and intraoperative variables were assessed for association to UPE by univariate analyses. Variables with a *p*-value of 0.1 or less were selected due to clinical relevance and included into the multivariable logistic regression analysis with backward elimination to assess their relative impact (adjusted odds ratio, OR) on the occurrence of UPE. Included variables were age, gender, preoperative creatinine (mg/dL), impaired left and right ventricular function, atrial fibrillation, pulmonary hypertension (>60 mm Hg) and intraoperative factors such as cardiopulmonary bypass, aortic clamp time and transfusion of red blood cell concentrates. The predictive value of the multivariable logistic regression model was estimated using the Hosmer–Lemeshow χ^2^ test.

Focusing on the impact of UPE, we adjusted the survival analysis for potential confounders by Cox regression. Pre-, intra- and postoperative prognostic factors for mortality during FU were first identified by Cox regression with forward selection based on the likelihood ratio test and incorporated into the final Cox regression model when *p* was ≤0.05. Included variables were UPE, gender, age ≥70 years, impaired RVF, COPD, drainage loss ≥800 mL and length of ventilation ≥48 h. 

The association between the percentage of patients with UPE and the year of surgery was analyzed by Spearman’s correlation. Missing values were excluded pairwise and are shown in Table 1, Table 2 and Table 3 when exceeding 5%. All tests were conducted two-tailed with a significance level of *p* ≤ 5%. Data were analyzed with IBM SPSS Statistics for Windows (Version 26.0, IBM Corp., Armonk, NY, USA.

## 6. Results

### 6.1. Baseline and Perioperative Patient Characteristics

The main baseline findings are summarized in Table 1. No differences in outcome were observed between the different surgeons (*p* = 0.61). Seventy-two of 393 (18.3%) patients showed X-ray-proven right-side UPE 24 h after surgery. The percentage of patients who developed UPE showed a slightly decreasing trend of the analyzed years (Spearman’s rank correlation coefficient −0.553, Figure 2), however, without reaching statistical significance (*p* = 0.097). Patients in the UPE group showed a significantly worse preoperative creatinine level compared to that in non-UPE patients, (1.03 vs. 0.93, *p* = 0.002, Figure 3). Patients with non-UPE presented more often with a good left ventricular ejection fraction (LVEF) compared to patients who developed UPE (82.2 vs. 71.8%, *p* = 0.047), while moderate LVEF was observed slightly more frequently in the UPE group (26.8 vs. 16.9%, *p* = 0.053). Patients in the non-UPE group presented more often with impaired right ventricular function, however, without reaching statistical significance. Mitral valve reconstruction was performed in 77.7%, and necessary mitral valve replacement in 22.3% of the patients. In total, *n* = 38 (9.8%) patients received closure of a patent PFO with significantly more PFO closures performed in the UPE group than in the non-UPE group (UPE = 16.7% vs. non-UPE = 8.3%; *p* = 0.031). Patients with UPE presented more often with atrial fibrillation (47.2 vs. 33.3%, *p* = 0.027). The use of concomitant tricuspid vale repair and cryo-Maze procedures tended to be more often in the UPE group but revealed no significant difference (*p* = 0.081).

C-reactive protein (CRP) was slightly higher in the UPE group but likewise without statistical significance. Receiver operating characteristic (ROC) analysis revealed no threshold value for preoperative CRP to predict the risk for the development of UPE (AUC:0.538).

Intraoperatively, patients who developed UPE had significantly longer procedural-times (Figure 4) with a bypass time of 210 min compared to 189 min in the non-UPE group and a higher necessity for inotropics (82.9 vs. 70.3%, *p* = 0.033) and enoximone (56.7 vs. 52.4%, *p* = 0.042, Table 2). Indicating a right heart burden, a higher percentage of patients required milrinone in the UPE group compared to the non-UPE group. 

Patients in the UPE group required mechanical ventilation significantly longer compared to the non-UPE patients (1110 vs. 820 min, *p* < 0.001), and tracheostomy was necessary more often (15.3 vs. 3.4%, *p* = 0.001). After mechanical ventilation, a higher portion of patients in the UPE group needed non-invasive-ventilation (10.0 vs. 3.2%, *p* = 0.022). Total drainage loss (975 vs. 500 mL, *p* < 0.001) and the maximum CRP value within 48 h postoperative were significantly higher in the UPE group vs. the non-UPE group (17.6 vs. 14.9, *p* = 0.007). No significant difference regarding volume management or liquid balance was found when comparing both groups. The Horovitz Index at ICU admission was significantly lower in the UPE group (390 vs. 475, *p* < 0.001). ICU length of stay (3 vs. 2 d, *p* < 0.001) and in hospital stay (13 vs.11d, *p* = 0.015) were significantly longer in the UPE group. Of MIMVS patients, 1.8% died during their hospital stay, and no significant difference with regard to 30-day mortality was found between the patient groups (*p* = 0.615). The combined in-hospital and 30-day mortality was higher in the UPE group compared to the non-UPE group (4.2 vs. 1.6%), but this finding did not reach significance (*p* = 0.165).

### 6.2. Patients after MIMVS on ECLS or ECMO Support

Six patients (8.3%) who developed UPE required an extracorporeal life-support (ECLS) system due to severe impairment of general oxygenation and low-cardiac output after MIMVS (Table 3). It is notable that in 5/6 patients, a closure of a patent foramen ovale was performed. One patient switched from ECLS-to vv-ECMO after 3 days of ECLS support due to hemodynamic recovery. ECLS therapy started 76.3 ± 94.3 min after MIMVS. Mean duration of ECLS therapy was 257.5 ± 91.8 h and a maximum of 167 h for vv-ECMO support. Of these patients, only one died of severe cerebral ischemia five days after surgery. In the ICU, five patients received a dilatation tracheotomy and four patients developed renal failure with hemodialysis. Re-thoracotomy was necessary in four patients, whereas two patients received a laparotomy because of intestinal ischemia, and one patient suffered from liver failure resulting from ischemic cholangitis. 

### 6.3. Survival Analysis

The median follow-up time of MIMVS patients was 3.9 years and did not differ between UPE and non-UPE patients (*p* = 0.947). Overall survival of MIMVS patients after 3, 5 and 9 years was 93, 88 and 77%, respectively (Figure 5A). No difference regarding survival was seen between UPE and non-UPE patients (*p* = 0.978, Figure 5B). Survival after 5 years was 89% in UPE patients and 88% in non-UPE patients. Patients of the UPE group who required ECMO or ECLS showed a significantly lower survival rate (*p* = 0.027), and none survived to the 10-year follow-up (Figure 5C). 

### 6.4. Logistic Regression Analysis

Multivariable logistic regression analysis identified five independent risk factors for the development of a UPE after MIMVS (Table 4); creatinine (OR 2.588), LVEF ≤ 50% (OR 2.074), pulmonary hypertension (OR 2.522), aortic clamp time (OR 1.009) and transfusion of red blood cell concentrate (OR 1.795). Impaired right ventricular function showed a protective effect (OR 0.339). Gender and age did not prove to be risk factors for UPE in our analysis. The model fit based on the Hosmer–Lemeshow χ^2^-test was 0.114.

### 6.5. Cox Regression Analysis

Independent risk factors for long-term mortality were age ≥ 70 years, impaired RVF, COPD, total drainage loss ≥800 mL and total length of ventilation ≥48 h. We could not prove a relation of UPE to survival after adjustment for confounding factors and also not for ECMO/ECLS or gender (Table 5). 

## 7. Discussion

Over a 10-year period, our study revealed no significant differences regarding the combined 30-day and in-hospital mortality after MIMVS for UPE and non-UPE patients (*p* = 0.165). The overall incidence of UPE was 18.3% (72/393 patients) with a slightly lower prevalence over 10 years in our patient cohort (Figure 5).

In addition, no difference regarding survival was seen between UPE and non-UPE patients (*p* = 0.98) after a median follow-up time of 4.6 years. Six patients with UPE (6/70, 8.6%) required ECMO/ECLS support due to severe impairment of general oxygenation in combination with a significantly worse long-term outcome (*p* = 0.027). Additional closure of a patent foramen ovale occurred significantly often in the UPE patients (*p* = 0.03).

Interestingly, multivariable logistic regression analysis identified preoperatively elevated creatinine values, reduced left ventricular function (<50%), pulmonary hypertension, increased aortic clamping time and transfusion RBCs as independent risk factors for the development of UPE after MIMVS. However, survival analyses of 386 patients after MIVMS estimated an excellent overall survival of 88% after 5 years and of 77% after 10 years. Independent risk factors for long-term mortality were aged ≥70 years, impaired RVF, COPD, total drainage loss ≥800 mL and total length of ventilation ≥48 h, but not UPE, ECMO/ECLS or gender.

In general, MIMVS has become a widely accepted standard procedure for isolated MIMVS. Many studies indicate that MIMVS produce comparable results for complex mitral valve repair with its limitations and longer procedural times [5,14]. Our 30-day survival rate of 98.2% and 10-year survival rate of 77% is in line with the latter studies and demonstrates that MIMVS can be performed as safely as a standard sternotomy mitral therapy with a repair rate of 77%. 

We further demonstrated that the incidence of UPE remains stable with a slightly decreasing tendency over a 10-year period reflecting higher expertise over the observation period, without having a significantly worse outcome in comparison to non-UPE patients in the short- and long-term follow-up (*p* = 0.978). One reason for this might be that the already known confounding factors such as COPD, total length of ventilation and high drainage loss have a higher impact on the long-term outcome than UPE itself.

Especially for MIMVS, UPE has been considered a unique complication for many years with different incidences [6,7,8,9,15]. UPE so far remains an unsolved phenomenon, though many therapeutic options have been discussed over the last years, and several working groups have evaluated this complication on small sample sizes [15,16].

The incidence of UPE in our study population in fact is higher than in other studies. Irsawara and coworkers reported only 2.1% of patients with UPE [15]. Tutschka and colleagues found a higher proportion of 25% as we did [8]. In the treatment study by Keyl and coworkers, only 7.9% of patients with UPE were observed in 2015. Of all UPE patients treated with dexamethasone, only one patient was clinically symptomatic. Their logistic regression analysis showed four variables associated with the development of UPE: dexamethasone, diabetes mellitus, the level of mean pulmonary arterial pressure and transfusion of fresh frozen plasma [7]. Fittingly, we found that the intraoperative administration of RBC is associated with the incidence of UPE. Transfusion requirements were found to be significantly different between patients with and without UPE in our analysis. Especially, RBCs have been associated with an increased inflammatory response and a proven increased risk of lung injury after cardiac surgery [17]. The varying incidence of UPE may be due to different diagnostic criteria, as well as to differences in the patient cohorts and perioperative management and should be discussed further.

Our definition of UPE was firstly clinically determined by the decline of the patient´s Horowitz Index, indicating deteriorated lung function and secondly by additional chest radiographs up to 24 h later. In accordance with Kesävuori and coworkers, we demonstrated that in general, the Horowitz Index for UPE patients was significantly worse upon ICU admittance (*p* = 0.001). This indicates functional impairment of the lungs in the early postoperative stage, especially in the group of postoperative UPE [10]. 

The intraoperative ventilation parameter such as tidal volume (*p* = 0.696), PEEP (*p* = 0.490) and FIO2 (*p* = 0.463) were not significantly different in our patient cohort. The disparity between the UPE and non-UPE group regarding perfusion and ventilation may explain the initial significant difference in the Horovitz Index in our study. Though there was no difference between the postoperative ventilation parameters for the PEEP and tidal volume and additionally for none of the intravenous fluid management between UPE and non-UPE in our patient cohort, a further improvement of the lung function is to be assumed without any further course of specific therapy. 

Though only one case report exists that discusses how one-lung ventilation induces re-expansion pulmonary edema (RPE) after MIMVS [6], it was demonstrated in an animal study by Leite and coworkers that RPE after bronchial occlusion evokes an acute lung and consecutive systemic inflammatory response by interleukins and cytokines [18]. Inoue and coworkers introduced a preventive protocol for re-expansion pulmonary edema during minimal invasive mitral valve surgery that consists of intermittent ventilation of the right lung, restoration of bilateral ventilation, administration of mannitol before unclamping the aorta and institution of mild hypothermia. By introducing this protocol, the incidence of the latter decreased significantly [16]. Kesävuori and colleagues already showed in another study that a one lung “minimal ventilation” concept appears to be beneficial in terms of postoperative total ventilation time and blood lactatemia in robotically assisted cardiac surgery, while there was no improvement in arterial blood gas measurements or in the rate of UPE [19]. Along with the described protective protocols, we tried to avoid prolonged sole single lung ventilation in close communication with the anesthetist, too.

However, besides the clinical signs of lung impairment, we might have observed some more radiographically mild cases of UPE in our analysis by classifying UPE according to our further published study [9]. In contrast to radiographs, a thoracic CT scan as suggested by Baik and coworkers would allow for a distinct evaluation of the affected lung for re-expansion pulmonary edema and would have been more favorable [20]. 

In a recent publication, Kesävuori and coworkers tried to classify the degree of UPE and investigated 231 patients after robotically assisted elective MIMVS. The chest radiographs were divided into three UPE grades based on the severity of radiological signs of pulmonary edema. They concluded that median total ventilation times were significantly longer with increasing severity of UPE [10]. Fittingly, the significantly longer ventilation time in the UPE in comparison to the non-UPE cohort was investigated in our study.

Concomitant procedures might increase the CPB time, and it is additionally notable that in 5/6 of the severe UPE cases in need of ECMO/ECLS, a patent foramen ovale was surgically closed. Normally, the blood flow of a patent foramen ovale ranges between approximately 50 and 300 mL/min/kg and correlates inversely with pulmonary arterial blood flow [21]. After closure, one might speculate that an increase in blood flow through the lungs in addition to the postoperative inflammatory response might aggravate the predisposition and occurrence of UPE. Though, the incidence of preoperative atrial fibrillation as a general risk factor for worse outcome was higher in the UPE cohort in our logistic regression analysis. Nevertheless, atrial fibrillation (AF) did not reach significance. The incidence of concomitant AF ablation procedures did tend to be more common in the UPE group but revealed no significance (*p* = 0.081). 

Along with the findings of many other studies [7,8,9,10,16], we confirmed that prolonged aortic clamping associated with CPB times is one of the major risk factors for UPE. We could not prove a significant change in CPB time over the course of operating years (*p* = 0.091); therefore, we assume a comparable impact of this risk factor over time. It is obvious that prolonged CPB times in general have some negative effects on the patient’s outcome [22]. These negative effects are generally accepted to be predominantly systemic; however, they may have an impact on the pathophysiology of the UPE. The general postprocedural inflammatory response might impair the re-expanded and re-perfused lungs that were collapsed during the MIMVS. This ischemia reperfusion injury might be aggravated by the generalized inflammatory response, promoted by CPB. Consequently, this mechanism may explain the association between prolonged cardiopulmonary bypass time with simultaneous prolonged lung collapse and unilateral lung injury [23,24]. Though we did not see any differences between the UPE and non-UPE study groups in temperature, ventilation of right lung and the access site, it is still unproven whether lifting of the pericardium during MIMVS restricts the backflow of blood of the pulmonary veins into the LA during the procedure or not. This issue has been already discussed by Renner and co-workers [9].

In contrast to the earlier analysis by Renner and coworkers, we could neither prove a correlation between the preoperative C-reactive protein and the incidence of the UPE nor define a threshold value for CRP predicting UPE (ROC analysis with AUC 0.538) [9]. Therefore, the theory that patients with postoperative UPE have a higher preoperative CRP indicating a higher disposition to a systemic inflammatory response is not evident in our analysis. In the current data analysis, the UPE patients have higher postoperative CRP values than the non-UPE patients (*p* = 0.007). This may strengthen the theory of a postoperative systemic inflammatory response provoked by the longer CPB times that aggravate local lung injury by an ischemia reperfusion injury of the lung.

## 8. Conclusions

To conclude, we demonstrated in our data that the outcome of patients after the occurrence of UPE was not significantly worse than in non-UPE patients over a 10-year time period (Figure 6), though a higher number of patients could lead to the detection of significant differences regarding outcome of UPE patients in general. By reducing the risk factors of UPE, the incidence could be reduced over the years. Despite ideal guideline-driven respiratory and intravenous volume therapy [25], we could not prevent UPE for patients over the ten years. 

We further demonstrated that besides the already known risk factors, other factors such as the transfusion of RBCs and closure of a patent foramen ovale were important risk factors for the development of a UPE in our patient cohort. It can be assumed that the combination of all already known risk factors increases the likelihood of severe UPE. Managing and minimizing preoperative risk factors and operative times may prevent UPE after MIMVS.

## 9. Limitations

The major limitation of this retrospective observational study is the potential impact of confounding factors in a non-randomized patient group. Therefore, we cannot show causal relations, but only generate hypotheses for further studies. The limited size of the UPE group reduces the statistical power and complicates the identification of significant differences for rarely occurring phenomenon and may influence the detection of differences between UPE and non-UPE patients according short- and long-term mortality, ICU and hospital stay. Furthermore, in case of the limited numbers of MIMVS surgery, a large experience with over 50 cases per year in each of the two surgeons who performed the minimally invasive mitral valve surgery is lacking. We cannot exclude that some more mild forms of UPE were investigated in using the inclusion criteria described by Renner and colleagues [9]. A graded classification of UPE by chest radiographs or CT may have allowed a more differentiated view of our results. 

## Figures and Tables

**Figure 1 jcm-10-02411-f001:**
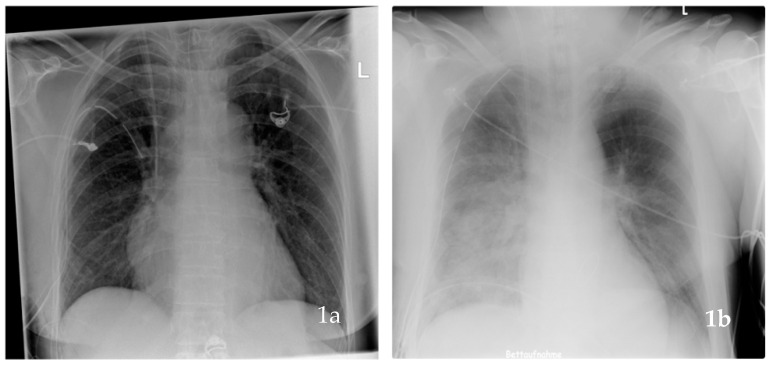
Two chest X-rays at admission to intensive-care-unit after minimal-invasive mitral-valve surgery. (**a**) chest X-ray showing normal lung tissue (no unilateral lung edema), with a chest tube and a central venous line in the upper vena cava. (**b**) chest X-ray of a patient with right sided unilateral pulmonary edema (UPE), i.e., more than 20% opafication of the right hemithorax and a chest tube and a central venous line in place.

**Figure 2 jcm-10-02411-f002:**
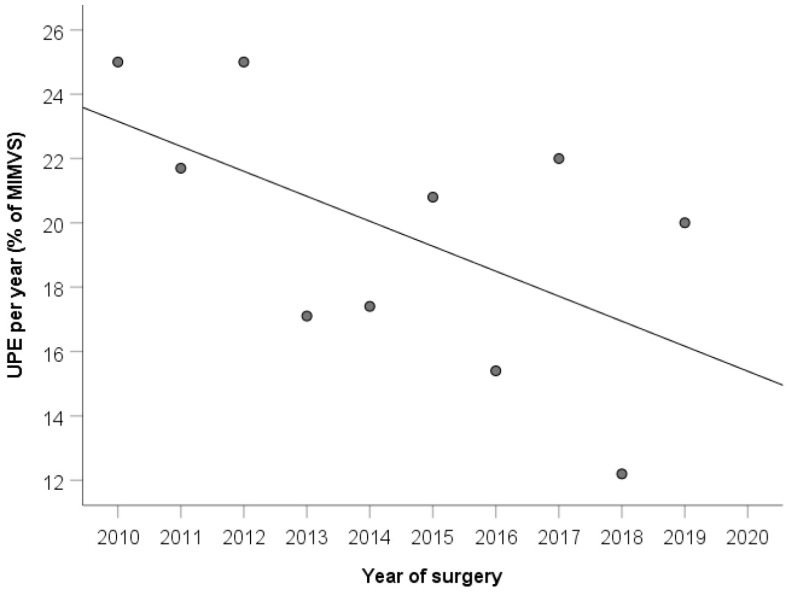
UPE per year (% of MIMVS). UPE, unilateral pulmonary edema; MIMVS, minimal-invasive mitral valve surgery.

**Figure 3 jcm-10-02411-f003:**
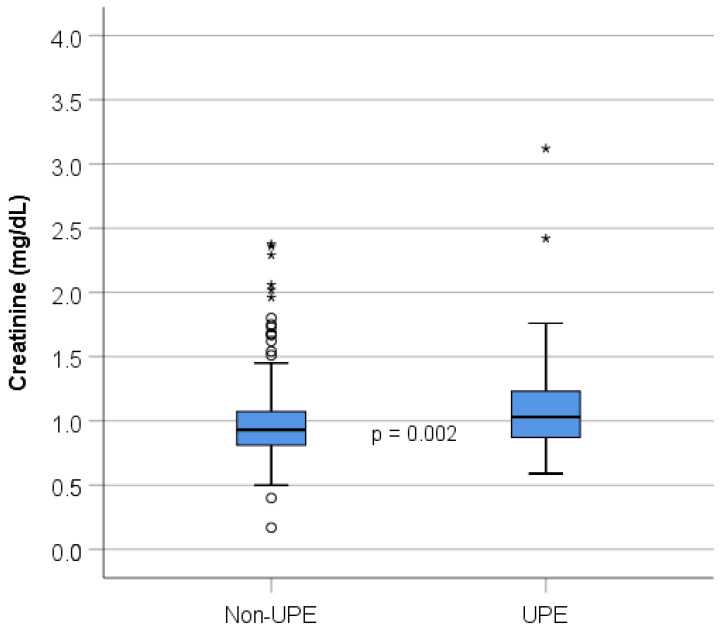
Creatinine levels in the non-UPE and the UPE groups. UPE, unilateral-pulmonary-edema; one outlier (non-UPE, 7.6 mg/dL) was excluded for the graphical representation. ◦ outlier; * extreme outlier.

**Figure 4 jcm-10-02411-f004:**
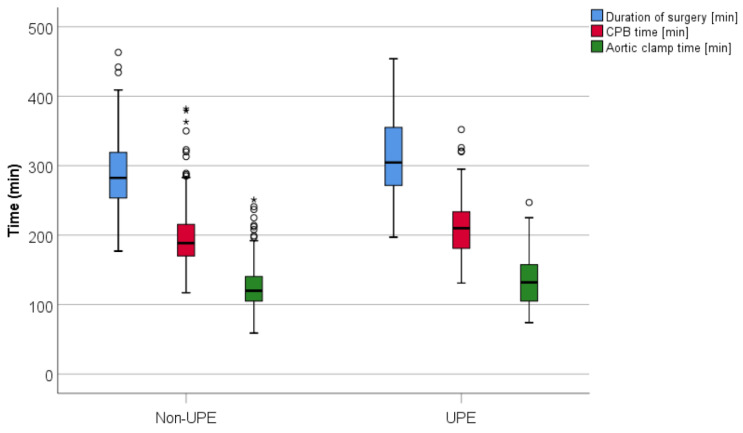
Duration of surgery, cardiopulmonary bypass (CPB) time and aortic clamp time in the non-UPE and the UPE groups. UPE, unilateral pulmonary edema; three outliers (duration of surgery, non-UPE, 590–1139 min) were excluded for the graphical representation, ◦ outlier; * extreme outlier.

**Figure 5 jcm-10-02411-f005:**
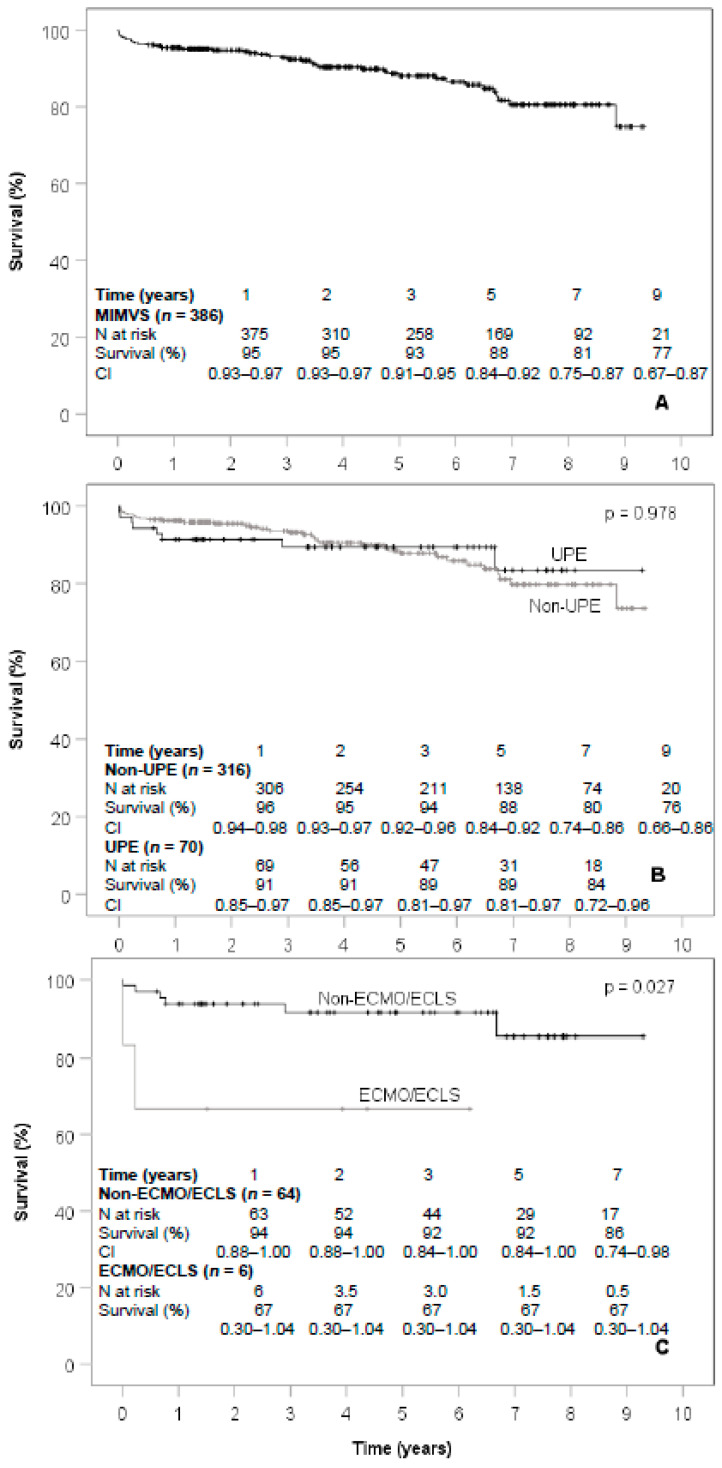
Kaplan-Maier long-term survival curves of the analyzed subgroups. (**A**) Survival curve of the MIMVS patients. (**B**) Survival curves of the UPE group and the non-UPE group. (**C**) Survival curves of patients requiring ECLS vs. non-ECLS within the UPE group. N, number; MIMVS, minimal-invasive mitral valve surgery; CI, confidence interval; UPE, unilateral pulmonary edema; ECLS, extracorporeal life support; ECMO extracorporeal membrane oxygenation.

**Figure 6 jcm-10-02411-f006:**
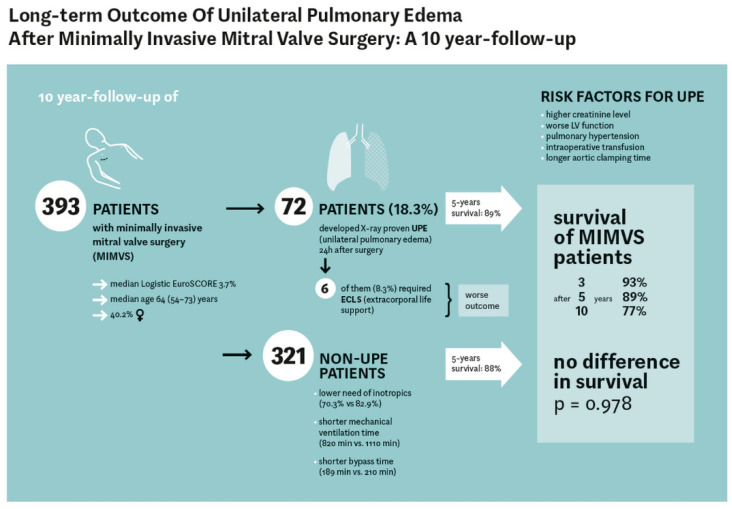
**shematic overview of the study:** Long-term outcomes of unilateral pulmonary edema after minimally invasive mitral 325 valve surgery: a 10-year follow-up.

**Table 1 jcm-10-02411-t001:** Preoperative patient details of MIMVS patients.

Variable	MIC (*n* = 393)	Non-UPE (*n* = 321)	UPE (*n* = 72)	*p*-Value
Age (years)	64 (54–73)	63 (54–73)	65.5 (53–73)	0.873
Female, *n* (%)	158 (40.2%)	129 (40.2%)	29 (40.3%)	0.989
Additive Euro I	5 (3–7)	5 (3–7)	5 (3–7)	0.229
Logistic EuroSCORE I	3.67 (2.08–6.59)	3.57 (2.08–6.30)	3.96 (2.27–7.35)	0.169
EuroSCORE II	0.98 (0.64–1.56)	0.95 (0.62–1.53)	1.10 (0.70–1.62)	0.182
Body mass index (kg/m^2^)	24.9 (22.7–27.7)	24.7 (22.4–27.8)	25.1 (24.0–27.6)	0.109
Creatinine (mg/dL)	0.95 (0.82–1.10)	0.93 (0.81–1.08)	1.03 (0.87–1.24)	0.002
Arterial hypertension	254 (65.3%)	205 (64.5%)	49 (69.0%)	0.467
IDDM	11 (2.8%)	10 (3.1%)	1 (1.4%)	0.697
Left ventricular ejection fraction ≤30%	4 (1.0%)	3 (0.9%)	1 (1.4%)	0.553
Impaired right ventricular function	75 (19.8%)	67 (21.5%)	8 (11.9%)	0.074
Sinus rhythm	257 (65.6%)	219 (68.4%)	38 (52.8%)	0.012
Atrial fibrillation	140 (35.9%)	106 (33.3%)	34 (47.2%)	0.027
Pacemaker	19 (4.9%)	16 (5.0%)	3 (4.2%)	1.000
COPD	36 (9.2%)	27 (8.5%)	9 (12.7%)	0.267
Smoking	79 (20.4%)	62 (19.6%)	17 (24.3%)	0.374
Active endocarditis	21 (5.5%)	16 (5.1%)	5 (7.0%)	0.561
Peripheral atrial disease	7 (1.8%)	5 (1.6%)	2 (2.9%)	0.616
TIA	5 (1.3%)	3 (1.0%)	2 (2.8%)	0.230
Stroke	12 (3.1%)	9 (2.9%)	3 (4.2%)	0.469
Previous cardiac surgery	9 (2.3%)	9 (2.9%)	0 (0.0%)	0.376
Previous mitral valve surgery	2 (0.5%)	2 (0.6%)	0 (0.0%)	1.000
CRP > 5 mg/dL preoperative	18 (4.6%)	14 (4.4%)	4 (5.6%)	0.754
CRP (mg/dL) preoperative	0.31 (0.13;1.19)	0.31 (0.12; 1.19)	0.44 (0.14;1.16)	0.337
Pulmonary hypertension (>60 mmHg)	58 (15.0%)	44 (13.9%)	14 (20.3%)	0.177

FFP = fresh frozen plasma, MIC = minimal-invasive surgery, UPE = unilateral reperfusion edema, IDDM = insulin dependent diabetes mellitus, COPD = chronic obstructive pulmonary disease, TIA = transient ischemic attack, CRP = C reactive protein.

**Table 2 jcm-10-02411-t002:** Intraoperative details of MIMVS patients.

Variable	MIC (*n* = 393)	Non-UPE (*n* = 321)	UPE (*n* = 72)	*p*-Value
Duration of surgery (min)	288 (258–323)	283 (254–320)	305 (271–355)	0.001
Bypass time (min)	193 (171–220)	189 (170–216)	210 (180–234)	0.005
Aortic clamp time (min)	122 (105–145)	120 (105–140)	132 (104–158)	0.031
Transfusion of red blood cell concentrates	147 (38.0%)	113 (35.6%)	34 (48.6%)	0.044
RBC, ∑ (mL)	600 (600–1200)	600 (600–1200)	600 (600–1200)	0.137
Transfusion of fresh frozen plasma	8 (2.1%)	5 (1.6%)	3 (4.3%)	0.161
FFP, ∑ (mL)	1200 (900–1200)	1200 (750–1200)	1200 (900–2400)	---
Transfusion of platelets	91 (23.5%)	69 (21.8%)	22 (31.4%)	0.084
platelets, ∑ (mL)	250 (250–250)	250 (250–250)	250 (250–250)	0.207
Inotropics	280 (72.5%)	222 (70.3%)	58 (82.9%)	0.033
Milrinon, *n* (%)	44 (11.4%)	32 (10.1%)	12 (17.1%)	0.095
Milrinon (mL/h), (range)	8.0 (6.0–8.0)	8.0 (6.0–8.0)	8.0 (8.0–8.0)	0.049
Enoximone (Perfan)	212 (54.8%)	166 (52.4%)	46 (65.7%)	0.042
Norepinephrine at the end of surgery (µg/kg/min)	0.05 (0.02–0.10)	0.05 (0.02–0.09)	0.05 (0.03–0.11)	0.249
Epinephrine, *n*	97 (25.1%)	75 (23.7%)	22 (31.4%)	0.175
Tidal volume (mL)	600 (525–683)	600 (525–683)	590 (525–670)	0.696
PEEP (cmH_2_O)	8 (6–10)	8 (6–10)	8 (6–10)	0.490
FiO₂	1.00 (0.8–1.0)	1.0 (0.8–1.0)	1.0 (0.8–1.0)	0.463
Mitral valve reconstruction	303 (77.7%)	248 (78.0%)	55 (76.4%)	0.769
Mitral valve replacement	87 (22.3%)	70 (22.0%)	17 (23.6%)	0.769
Cardiac valve size	29 (29–31)	29 (29–31)	29 (29–31)	0.824
Combined interventions				
PFO/ASD closure	38 (9.8%)	26 (8.3%)	12 (16.7%)	0.031
Cryo-MAZE procedure	6 (1.6%)	3 (1.0%)	3 (4.2%)	0.081
Tricuspid valve reconstruction	6 (1.6%)	3 (1.0%)	3 (4.2%)	0.081
Closure of the left atrial appendage	4 (1.0%)	2 (0.6%)	2 (2.8%)	0.160
Other	9 (2.3%)	7 (2.2%)	2 (2.8%)	0.677

∑ = sum, FFP = fresh-frozen-plasma, PEEP = positive-end-expiratory-pressure, PFO = persistent foramen-ovale, ASD = atrial-septal-defect.

**Table 3 jcm-10-02411-t003:** Postoperative details of MIMVS patients.

Variable	MIC (*n* = 393)	Non-UPE (*n* = 321)	UPE (*n* = 72)	*p*-Value
Re-thoracotomy	30 (7.7%)	21 (6.6%)	9 (12.5%)	0.088
Length of ventilation (min)	0.8 (0.6–640)	0.8 (0.5–654)	1.0 (0.7–571)	0.143
∑ Length of ventilation (min)	850 (660–1170)	820 (645–1050)	1110 (826–4034)	<0.001
NIV	17 (4.5%)	10 (3.2%)	7 (10.0%)	0.022
Tracheostomy	22 (5.6%)	11 (3.4%)	11 (15.3%)	0.001
ECMO	2 (0.5%)	1 (0.3%)	1 (1.4%)	0.335
ECLS, *n* (%)	6 (1.5%)	0 (0.0%)	6 (8.3%)	<0.001
Total drainage loss (mL)	550 (300–1350)	500 (300–1113)	975 (413–2375)	<0.001
CRP max 48 h postoperative (mg/dl)	15.2 (11.7–20.3)	14.9 (11.4–20.0)	17.6 (12.7–22.0)	0.007
Re-ITN, *n* (%)	27 (6.9%)	19 (6.0%)	8 (11.1%)	0.126
Horovitz Index at ICU arrival, 8.9% missing	460 (367–540)	475 (387–549)	390 (299–478)	<0.001
Horovitz Index (ICU < 4 h), 61.8% missing	330 (244–425)	335 (236–430)	315 (244–406)	0.746
Length of stay at the ICU (days)	2 (2–3)	2 (2–2)	3 (2–7)	<0.001
Length of stay at the hospital (days)	12 (9–15)	11 (9–14)	13 (10–19)	0.015
Combined in-hospital and 30-day mortality	8 (2.0%)	5 (1.6%)	3 (4.2%)	0.165
Follow-up time (years)	3.9 (1.6–6.4)	3.9 (1.6–6.3)	4.1 (1.5–6.5)	0.947

NIV = non-invasive ventilation, ECMO = extracorporeal-membrane-oxygenation, ECLS = extracorporeal-life-support, ITN = intubation, ICU = intensive-care-unit.

**Table 4 jcm-10-02411-t004:** Logistic regression analysis for the occurrence of UPE after MIC.

Predictors	Odds Ratio	95% CI	*p*-Value
Creatinine (mg/dL)	2.588	1.146–5.843	0.022
Impaired RVF	0.339	0.131–0.880	0.026
LVEF < 50	2.074	1.093–3.933	0.026
Pulmonary hypertension (>60 mm Hg)	2.522	1.072–5.931	0.034
Aortic clamp time (min)	1.009	1.001–1.017	0.037
Transfusion of red blood cells	1.795	1.028–3.137	0.040

**Table 5 jcm-10-02411-t005:** Cox regression analysis for long-term mortality risk factors after MIC.

Predictors	Hazard Ratio	95% CI	*p*-Value
UPE	0.810	0.357–1.838	0.615
Age ≥ 70 years	2.828	1.488–5.376	0.002
Impaired RVF	2.739	1.418–5.291	0.003
COPD	2.759	1.376–5.532	0.004
Total drainage loss ≥800 mL	2.993	1.248–7.174	0.014
∑ Length of ventilation ≥48 h	3.834	1.831–8.024	<0.001

## Data Availability

The data presented in this study are available on request from the corresponding author.

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
