# Peer review of "Outcome of Unilateral Pulmonary Edema after Minimal-Invasive Mitral Valve Surgery: 10-Year Follow-Up"

_jcm, 2021, doi:10.3390/jcm10112411_

Round 1

Reviewer 1 Report

I have read your study with great curiosity. The topic is significant as the study aims to answer crucial questions about complication rates and survival in cardiac surgery nowadays. Your research is well-designed, with a widely described methodology, sufficient study group, and a very long follow-up time. 
I have one minor question, why you decided to define pulmonary hypertension > 60 mmHg in your study, and how did you measure it? 
Thank you for your work. Overall this is a very thought-provoking and accurately conducted study. 

Author Response

Reviewer 1:

R1 Comment 1: I have read your study with great curiosity. The topic is significant as the study aims to answer crucial questions about complication rates and survival in cardiac surgery nowadays. Your research is well-designed, with a widely described methodology, sufficient study group, and a very long follow-up time. 
I have one minor question, why you decided to define pulmonary hypertension > 60 mmHg in your study, and how did you measure it? 
Thank you for your work. Overall this is a very thought-provoking and accurately conducted study.

AR1: Thank you for your positive feedback and your minor comment. The Pulmonary pressure was measured by echocardiography and right-heart catheterization. Pulmonary hypertension above and below 60mmHg was requested in our institutional database. We stated that in the Method part, line: 145, 146

Reviewer 2 Report

The manuscript carries important information but needs extensive revision:

1) Absence of evidence is not evidence of absence: patients with UPE carried a three fold higher mortality than non UPE patients. The fact that this was not statistically significant was due to the power of the study. This should be clearly stated and avoid to give the misleading impression in the conclusion section that the hospital mortality between the two groups are similar. The same applies to the ICU stay and  hospital stay.

2) There is no reference to the single lung ventilation duration which has been identified as risk factor from other studies. The anaesthetist and the  surgeon should make attempts to minimise and also interrupt the single lung  ventilation period, in order to lessen the risk of ULE. Was this the case with the anaesthetic and surgical management in the particular groups of patients?

3) The operation time, cross clamp time and CPB time are all prolonged for this type of surgery (ideally should be halved). Also the number of operations for a 10 year period is rather small (40 cases per year) if it is multiple surgeons experience. Is this a single surgeon or multiple surgeons experience? If it is a single surgeon then the numbers are sufficient. If it is a 4 or 5 surgeons experience, each one doing less than 10 a year, then the numbers are not sufficient to maintain a high level of skills.

4) Use the EuroSCORE2 instead of the logistic Euroscore in the abstract.

5) In hospital or 30 day mortality should be presented together and not separately.

6) I would advise the review of the paper by a medical statistician.

Author Response

Reviewer 2:

The manuscript carries important information but needs extensive revision:

R2 comment 1) Absence of evidence is not evidence of absence: patients with UPE carried a three-fold higher mortality than non UPE patients. The fact that this was not statistically significant was due to the power of the study. This should be clearly stated and avoid to give the misleading impression in the conclusion section that the hospital mortality between the two groups are similar. The same applies to the ICU stay and hospital stay.

AR2: Thank you for your comment. We followed your advice and stated in the Conclusion and Limitations section that the number of patients treated could lack evidence of worth outcome for the UPE patients: “To conclude, we demonstrated in our data that the outcome of patients after the occurrence of UPE was not significantly worse than in no-UPE patients over a 10-year time-period (Figure 5), though a higher number of patients could lead to the detection of significant differences regarding outcome of UPE patients in general.” (line 405-408). The limited size of the UPE-group reduces the statistical-power and complicates the identification of significant differences for rarely occurring phenomenon and may influence the detection of differences between UPE and non UPE patients according short and long-term mortality, ICU and hospital stay (line 420-423).

R2 comment 2) There is no reference to the single lung ventilation duration which has been identified as risk factor from other studies. The anesthetist and the surgeon should make attempts to minimise and also interrupt the single lung ventilation period, in order to lessen the risk of ULE. Was this the case with the anaesthetic and surgical management in the particular groups of patients?

AR2: As the reviewer pointed out correctly single lung ventilation has not only advantages for the surgical exposure, but also has the disadvantage of being a risk factor of lung edema, as described in one case report. Larger studies dealing with this topic are missing so far. Only preventive strategies exist that diminish single lung ventilation in addition to many other factors. As the reviewer pointed out right, we also try to minimize the sole single lung ventilation in close communication with the anesthetist.  This is now stated and discussed in the Discussion section: “Though only one case report exists, that one-lung-ventilation induces reexpansion-pulmonary-edema (RPE) after MIMVS (6) it was demonstrated in an animal study by Leite and coworkers, that RPE after bronchial occlusion evokes an acute lung and consecutive systemic inflammatory response by interleukines and cytokines (18). Inoue and coworkers introduced a preventive protocol for reexpansion pulmonary edema during minimal invasive mitral valve surgery that consists of intermittent ventilation of the right lung, restoration of bilateral ventilation, administration of mannitol before unclamping the aorta, and institution of mild hypothermia. By introducing this protocol, the incidence of the latter decreased significantly (19). Kesävuori and colleagues already showed in another study that a „minimal-ventilation “-concept appears to be beneficial in terms of postoperative total ventilation-time and blood-lactatemia in robotically-assisted cardiac-surgery while there was no improvement in arterial-blood gas measurements or in the rate of UPE (20). Along with the described protective protocols, we tried to avoid prolonged sole single lung ventilation in close communication with the anesthetist, too.” (line 342-355)

R2 Comment 3) The operation time, cross clamp time and CPB time are all prolonged for this type of surgery (ideally should be halved). Also the number of operations for a 10 year period is rather small (40 cases per year) if it is multiple surgeons experience. Is this a single surgeon or multiple surgeons experience? If it is a single surgeon then the numbers are sufficient. If it is a 4 or 5 surgeons experience, each one doing less than 10 a year, then the numbers are not sufficient to maintain a high level of skills.

AR2.  The Reviewer is right with his indication of a multiple surgeon’s experience: there has not been a large experience in each of the two surgeons who performed the minimally invasive mitral valve surgery. This is now added in the Limitations of the Discussion section: “Furthermore, in case of the limited numbers of MIMVS surgery, a large experience with over 50 cases per year in each of the two surgeons who performed the minimally invasive mitral valve surgery are lacking.” (line 424-426)

R2 comment 4. Use the EuroSCORE2 instead of the logistic Euroscore in the abstract

AR2: We thank the reviewer for this suggestion. We have replaced logistic Euroscore by EuroSCORE2 in the abstract: .  Median EuroSCORE II reached 0.98 in the complete MIMVS-group.  (line 76).

R2 comment 5. In hospital or 30-day mortality should be presented together and not separately.

AR2: We thank the reviewer for this important suggestion. We have replaced the separate mortalities by the combined mortality in table 3 and in the abstract lines 76-77, 82-83 and in the results section in lines 283-284.

R2 comment 6. I would advise the review of the paper by a medical statistician.

AR2.: We thank the reviewer for this important advice. We consulted a medical statistician and added a Cox regression analysis to our manuscript. We thank Dr. Sandra Freitag -Wolf for the critical revisions and discussions of the manuscript. We added further information to the statistic part: Focusing on the impact of UPE we adjusted the survival analysis for potential confounders by Cox regression. Pre- intra- and postoperative prognostic factors for mortality during FU were first identified by Cox regression with forward selection based on the likelihood ratio test and incorporated into the final Cox regression model when p was ≤ 0.05. Included variables were UPE, gender, age ≥ 70 years, impaired RVF, COPD, drainage-loss ≥ 800 ml and length of ventilation ≥ 48 hrs. (line 181-186) and Independent risk factors for long-term mortality were age ≥ 70 years, impaired RVF, COPD, total drainage-loss ≥ 800 ml and total length of ventilation ≥ 48 hours. We could not prove a relation of UPE to survival after adjustment for confounding factors, and also not for ECMO/ECLS or gender (line 274-278).

Reviewer 3 Report

The reviewer recommend authors to add Figure of survival curve which compares the frequency of in-hospital outcomes between patients with vs. without UPE.

Author Response

Reviewer 3:

The reviewer recommends authors to add Figure of survival curve which compares the frequency of in-hospital outcomes between patients with vs. without UPE.

AR3: We thank the reviewer for this valuable comment. We have performed an additional Cox-Regression analysis and included pre- and intraoperative factors as well as in-hospital outcomes to assess their adjusted impact on long-term-survival. We have summarized these findings in table 5, since we believe that this provides a more detailed information than an additional survival curve. The new findings are described in the abstract, methods, results and discussion sections. (line 82-84, 183-188, 274-278, 297-299, 309-311)